# Whole-Genome Sequencing Improves the Diagnosis of *DFNB1* Monoallelic Patients

**DOI:** 10.3390/genes12081267

**Published:** 2021-08-19

**Authors:** Anaïs Le Nabec, Mégane Collobert, Cédric Le Maréchal, Rémi Marianowski, Claude Férec, Stéphanie Moisan

**Affiliations:** 1Univ Brest, Inserm, EFS, UMR 1078, GGB, F-29200 Brest, France; megane_collobert@hotmail.fr (M.C.); cedric.lemarechal@univ-brest.fr (C.L.M.); claude.ferec@gmail.com (C.F.); 2Laboratoire de Génétique Moléculaire et d’Histocompatibilité, CHRU Brest, UMR 1078, F-29200 Brest, France; 3Service ORL et Chirurgie Cervicofaciale du CHRU Brest, F-29200 Brest, France; remi.marianowski@chu-brest.fr

**Keywords:** WGS, *DFNB1*, deafness, variants, functional assays, cCRE, non-coding variation

## Abstract

Hearing loss is the most common sensory defect, due in most cases to a genetic origin. Variants in the *GJB2* gene are responsible for up to 30% of non-syndromic hearing loss. Today, several deafness genotypes remain incomplete, confronting us with a diagnostic deadlock. In this study, whole-genome sequencing (WGS) was performed on 10 *DFNB1* patients with incomplete genotypes. New variations on *GJB2* were identified for four patients. Functional assays were realized to explore the function of one of them in the *GJB2* promoter and confirm its impact on *GJB2* expression. Thus, in this study WGS resolved patient genotypes, thus unlocking diagnosis. WGS afforded progress and bridged some gaps in our research.

## 1. Introduction

Hearing loss is the most common sensory pathology, affecting about 1–2 in every 1000 newborns, with a prevalence which increases with age [1,2,3,4]. In industrialized countries, congenital deafness has a genetic origin in 80% of cases [5]. Deafness can be syndromic or not (associated or not with other pathologies or malformations), respectively representing 10% and 90% of cases. More than 500 syndromes are associated with syndromic deafness and more than 100 genes have been described in non-syndromic hearing loss (NSHL) (Van Camp G, Smith RJH. Hereditary Hearing Loss Homepage. https://hereditaryhearingloss.org, the 10 May 2021).

NSHL can be classified according to heredity. Generally 80–90% of those affected have autosomal recessive inheritance (DFNB), 10–15% have a dominant mode (DFNA), 1% of cases are associated with the X chromosome (DFNX), and others have a mitochondrial inheritance mode [5,6].

The predominant form is autosomal recessive non-syndromic hearing loss (*DFNB1*). Most *DFNB1* phenotypes are described as prelingual and bilateral non-syndromic hearing loss, this being severe to profound. This type of deafness affects all frequencies and is not associated with inner ear malformations. Vestibular function remains unaffected [1,2,3,4,5,7,8]. The *GJB2* (Gap Junction β 2-chr13:20,187,470–20,192,938 (hg38)) gene is mainly implicated in *DFNB1* with frequencies ranging from 20% to 40%, according to populations with the most frequent mutation, c.35delG [3,5,9,10,11].

Moreover, seven large *DFNB1* deletions have been described in *DFNB1* patients: del-920 kb [12], del-101 kb del(*GJB2*-D13S175) [13], del(*GJB6*-D13S1830) [14], del(*GJB6*-D13S1854) [15], del-131kb [16], del-179kb [17], and del-8kb [18]. This year, Brozkova et al. described a *DFNB1* deletion of 3 kb in one patient [11].

The genomic architecture of our chromosomes is now far better investigated and understood. Many studies have focused on the role of non-coding regions and on genetic variants they contain, opening up new research possibilities. However, a vast majority of coding and non-coding variants may remain of unknown clinical significance [19,20,21].

Almost 8% of the human genome is covered with candidate *cis*-regulatory elements (cCREs) [22]. The identification of distal acting regulatory elements has been the object of active research in recent years. Disruptions of such regulatory elements and/or chromatin conformation are likely to play a critical role in human genetic diseases [19,20,21].

Routine molecular diagnosis in the Molecular Genetics Laboratory at Brest University Hospital involves the testing of around 80 deaf patients each year, and among these patients, ~20% are *DFNB1* biallelic carriers. However, several genotypes remain incomplete; for monoallelic *DFNB1*, which represents fewer than 1% of the tested patients, most patients are carriers of the c.35delG heterozygous and some have rare variants.

The c.35delG heterozygous genotype may be related to the general population frequency with an overall frequency of 2% but an excess of the deletion has been shown in the deaf population [8].

To accelerate a patient’s diagnosis odyssey, we propose the investigation of non-coding DNA such as *GJB2* CREs. Exploration of structural variations which could disrupt *DFNB1* 3D regulating looping model will be important. Moreover, variations in other deafness genes could explain phenotypes.

To resolve these cases, whole-genome sequencing (WGS) was performed on 10 monoallelic *DFNB1* carriers of rare variants. In this study, WGS resolved some genotypes, thus unlocking diagnoses. Moreover, we identified a new mutation in the *GJB2* promoter which impacts *GJB2* gene regulation.

## 2. Materials and Methods

### 2.1. Ethics Statement

All patients gave their informed consent for inclusion before they participated in the study. The study was conducted in accordance with the Declaration of Helsinki (1975), and the protocol was approved by the Ethics Committee of Brest (Protocol N° 29BRC19.0104).

### 2.2. Recruitment of Patient/Population

Ten monoallelic *DFNB1* patients with non-syndromic bilateral, stable, mild to profound deafness were included in this study after routine molecular diagnosis at the Molecular Genetics Laboratory at Brest University Hospital between 1990 and 2020. Eight patients with rare variant were diagnosed, including one patient (patient P5) with del(*GJB6*-D13S1830), and one c.35delG family case (patient P4). All patients had permanent hearing loss, not associated with infection or drugs. For these patients, CGH array genotyping had already been performed, with no deletion or duplication at the *DFNB1* locus detected. Before WGS, Sanger sequencing had been realized to screen exon 2 of the *GJB2* gene.

### 2.3. Whole Genome Sequencing

#### 2.3.1. Laboratory

DNA was submitted for whole-genome sequencing (Integragen Genomics platform). PCR free libraries were prepared with NEBNext Ultra II DNA Library Prep Kits according to supplier recommendations. Specific double-strand gDNA quantification and a fragmentation (300 ng of input with high-molecular-weight gDNA) sonication method were used to obtain approximately 400 bp fragments. Finally, paired-end adaptor oligonucleotides (xGen TS-LT Adapter Duplexes from IDT) were ligated and re-paired. Tailed fragments were purified for direct sequencing without a PCR step.

DNA PCR free libraries were sequenced on paired-end 150 pb runs on the Illumina NovaSeq apparatus. Image analysis and base calling were performed using Illumina Real Time Analysis (RTA) Pipeline version 3.4.4 with default parameters.

#### 2.3.2. Bioinformatics

Sequence reads were mapped on the Human Genome Build (hg38) with the Burrows-Wheeler Aligner (BWA) tool [23].

Integragen Genomics uses Broad Institute’s GATK Haplotype Caller GVCF tool (GATK 3.8.1) [24] for single nucleotide variations (SNV) and small insertions and deletions.

Variants were annotated with Ensembl’s VEP (Variant Effect Predictor, release VEP 95.1) [25] (which takes data available in gnomAD, the 1000 Genomes Project, Kaviar Databases…) by Integragen Genomics. Then, 5 algorithms (DANN, FATHMM, MutationTaster, SIFT and Polyphen) were used to predict pathogenicity of single nucleotide polymorphism (SNP) [26,27,28].

### 2.4. Data Analysis

#### 2.4.1. SNV Identification

Different analyses were performed: all deafness genes were annotated and variations with a frequency of less than 0.1 were analyzed (215 genes in Appendix A). Variations in all genes with a frequency of less 0.07% were also studied in public databases (1000 Genomes Project, GnomAD, Kaviar) [29].

#### 2.4.2. SV Identification

Integragen Genomics used different algorithms to investigate structural variations: Lumpy (v 0.2.13), Delly (v 0.7.9) and Manta (v 1.5.0) on bam files.

The BreakDancer algorithm was also used to investigate structural variations.

### 2.5. Confirmation and Segregation Analysis

#### Variant Confirmation

Each candidate variant was confirmed using Sanger sequencing on ABI3130XL (Thermo Fisher Scientific Inc., Waltham, MA, USA) using the Big Dye Terminator Cycle Sequencing V3.1 Ready Reaction Kit (Life Technologies, Carlsbad, CA, USA). Segregations were performed when the DNA of parents was available.

### 2.6. Functional Assays Cells

#### 2.6.1. Plasmid Constructs

All the cloning steps were done using the “In fusion^®^” strategy from Clontech. Using the pGL3-Basic Vector (Promega), the 5′-flanking region of the *GJB2* gene (1541 bp, “P*_GJB2_*”) was cloned upstream from the firefly luciferase cDNA at the Hind III site. CREs and combinations were amplified and inserted downstream. All the inserted fragments were verified by sequencing. The PCR primers used to amplify the *GJB2* promoter and CREs are given in Appendix A.

#### 2.6.2. Mutagenesis

Mutagenesis was performed using a QuickChange II XL Site-Directed Mutagenesis kit from Agilent Technologies. The primers are presented in Appendix A.

#### 2.6.3. Luciferase Assays

For luciferase assays, 1.25 × 10^5^ cells (SAEC: Small Airways Epithelial cells) were seeded in 12-well plates. Transfections were undertaken 24 h later with the transit 2020 reagent (Mirus). Here, 400 ng of the P*_GJB2_* constructs and 100 ng of a pCMV-LacZ construct (as an internal control) were used for each condition. Every condition was used in triplicate. At 48 h post-transfection the cells were washed once with 1× PBS and lysed with Passive lysis buffer (Promega). Cells lysates were clarified by centrifugation at 12,000× *g* for 5 min at 4 °C. Then, 20 µL of each protein extract was used to assay the luciferase activity and 25 µL for β-galactosidase activity. We used Promega reagents and the Varioscan multiwell plate reader (Thermo Fisher). Results are presented as relative luciferase activity, with the P*_GJB2_* construct activity equal to 1. The significance of the increased luciferase activity was determined using analysis of variance and t-tests with R.

## 3. Results

In order to detect an unknown causal variant, WGS was performed on 10 monoallelic *DFNB1* patients. This study involved nine patients who carried one *GJB2* heterozygous mutation (Table 1) and one patient (Patient P5) who carried the heterozygous deletion del(GJB6-D13S1830).

### 3.1. SNV

#### 3.1.1. *GJB2* Mutations

WGS analysis identified a second mutation on the *GJB2* gene in patients P3, P4, P8, and P10 (Table 2).

In the routine molecular diagnosis, the Sanger sequencing of the *GJB2* gene of patient P3 allowed us to detect frameshift variation (rs730880338) at c.269. Then, a missense variation (rs80338945) was discovered by WGS analysis at the same position. Indeed, these variations in the same nucleotide complicated the interpretation of Sanger sequencing analysis. The duplication hid the other mutation so that the initial Sanger sequencing interpretation failed to detect the two mutations.

DNA samples from parents were not available for segregation analysis, but single-strand NGS sequencing (see IGV (Integrative Genome Viewer) BAM visualization (Figure 1) confirmed that these variants are in *trans*.

Patient 4 carried two *GJB2* mutations, a recurrent mutation, in deaf population, only the c.35delG (rs8033893) have been detected by DHPLC (Denaturing High Performance Liquid Chromatography). The second mutation discovered by WGS analysis was a frameshift (rs730880338), c.269dup (Figure 2). Indeed, since no hetero- and homo-duplex was detected by HPLC we did not realize Sanger sequencing. This is why genotype was unresolved before WGS. Variant segregation of parents allowed the determination of variants transmission for their children (Figure 3).

The first Sanger sequencing of Patient P8 was performed in 2002; only the nonsense mutation of *GJB2* was detected (rs104894398—c.139G > T) (Figure 4).

WGS analysis allowed identification of a second *GJB2* mutation, a splice site mutation (rs80338940—c.-23 + 1G > A) on intron 1 of the *GJB2* gene (Figure 4). This variant had not been found earlier because the sequencing of exon 1 has only been routinely done in the laboratory since 2005.

After WGS, Sanger sequencing confirmed this second mutation, and segregation confirmed that these mutations are in *trans* (Figure 5). The genotypes of patient P8 and her brother are c.[-23 + 1G > A];[139G > T] p.[?];[Glu47Ter] (Figure 5).

#### 3.1.2. *GJB2* Upstream Variation

Patient P10 carried one nonsense *GJB2* mutation (c.139G > A;p.Glu47Ter). WGS analysis allowed us to identify two variants on *GJB2*: rs1425012952 and rs372782198. These variants were respectively upstream and downstream of *GJB2*. The upstream variant was located on the *GJB2* promoter, particularly on a GC box located at −81 bp of the TSS (Transcription Start Site) (Figure 6). This GC box has been described as useful and critical for *GJB2* basal transcription with the binding of Sp1 transcription factor [20]. This variant is reported once in GnomAD, with one carrier.

Sanger analysis of parents confirmed a *trans* segregation (Figure 7), indeed the mother was carrying this upstream *GJB2* variant, and the father the nonsense mutation. The upstream *GJB2* variant was not detected during routine molecular diagnosis the first time, because the sequencing concerned only exons and intron of *GJB2*.

Patient P10 carried these two mutations, a nonsense and an upstream variant, in *trans*, which may explain his deafness. To understand, the impact of *GJB2* upstream variant on *GJB2* gene expression, functional in-vitro assays were performed. Indeed, Contra v3 (in silico this is to predict transcription factor binding sites (TFBSs)) predicted several Sp1 TFBSs just at the location of this upstream variant (Figure 6). Moreover, in 1997, Kiang and et al., described *GJB2* regulation by binding of Sp1 transcription factor in the GC box -81.

Functional assays allowed us to analyze the *GJB2* promoter variant on *GJB2* gene expression by firefly luciferase expression following transient transfection into cells. For this, different constructions are realized, one with the WT *GJB2* promoter (P*_GJB2_*), one with the insertion of this variant on the *GJB2* promoter (P*_GJB2_* Δ). Moreover, in our previous paper, the C3 region (chr13:20419404-20420446 (hg38)—1043 pb) was described for the first time as the most important *GJB2* enhancer [30]. Thus, we also decided to study the impact of the upstream *GJB2* variant on *GJB2* expression with and without the C3 enhancer.

P*_GJB2_* WT construction has an expression equal to 1 and P*_GJB2_* Δ construction significantly decreased the *GJB2* expression (decrease of 17%). Moreover, the *GJB2* promoter with C3 enhancer (P*_GJB2_* WT + C3) showed an *GJB2* expression of 7 but this construction with the SNP promoter (P*_GJB2_* Δ + C3) exhibited a two-fold decrease in *GJB2* expression. (Figure 8).

#### 3.1.3. SNVs on Deafness Genes

For the five other patients carrying one nonsense mutation, no mutation was identified on *GJB2*, or on CREs of the *GJB2* gene and *DFNB1* locus. For this reason, we screened genes linked to deafness to identify potential pathogenic variations. Heterozygous pathogenic or likely pathogenic variants (single nucleotide, indel, or copy number variants) were detected in known NSHL genes in five probands but they were not sufficient to explain the phenotype.

##### USH1C Gene

Patient P5 was the only patient to carry a *DFNB1* deletion: del(*GJB6*-D13S1830). This deletion of 309 kb removed *CRYL1* and *GJB6* genes but *GJB2* remained intact (Figure 9). To explore any possibilities, we screened other deafness genes.

Among deafness genes, three variants have been identified on the *USH1C* gene: a nonsense variation (rs377145777), a missense variant (rs1064074), and a splice site variation with unknown impact on protein (Table 3). Parental DNA were not available to study segregation.

### 3.2. Structural Variations

Structural variations (SVs) corresponded to chromosomic rearrangements such as deletion, insertion, duplication, and translocation which could disrupt one or several loci, genes, or regulatory regions at the same time.

#### 3.2.1. CNV by Integragen Genomics

Integragen Genomics proposed a CNV (Copy Number Variation) study for all patients. This included only deletion and duplication variations. These analyses allowed us to confirm the deletion del(*GJB6*-D13S1830) of patient P5 (Figure 9). No other variant on other regions of the *DFNB1* has been detected by this tool for the other patients.

#### 3.2.2. BreakDancer Algorithm for SVs

To continue SV analysis, BreakDancer algorithm was used on all samples. The algorithm brought to light a few hundred variations by chromosome. We focused on chromosome 13 at the *DFNB1* locus implicated on NSHL.

For patients with unresolved genotypes, some SVs were detected on chromosome 13, around the *DFNB1* locus. These SV do not disrupt either the *GJB2* gene or the *DFNB1* locus but we do not know their impact on other genes.

## 4. Discussion

Hearing loss is the main sensorial deficit in developed countries, with more than 80% of cases being of genetic origin. Although more than 100 genes have been identified in non-syndromic deafness, for certain patients no genotypes nor environmental causes can be identified [5].

The key message of this work is the impact of WGS to improve molecular base knowledge of NSHL.

WGS analysis resolved genotypes for four patients. Indeed, for three patients, genotypes have been elucidated with a second mutation on *GJB2* gene. Indeed, for these patients the second event in *GJB2* has not been detected in routine care. Although, Whole Exome Sequencing (WES) could have been detected these variants on *GJB2* (as c.269T > C variant (P3), c.269dup (P4) c.-23 + 1G > A (P8)), WES would not have detected *GJB2* promoter variant (P10). Here, WGS allowed for the study of non-coding and intronic DNA besides coding sequences. Thus, WGS allows the detection of small deletion in the non-coding regions of the genome as well as some inversion that is not detected by Exome sequencing. Moreover, the cost of WGS is close to the cost of WES, and will probably decrease in the next years.

For one patient, two variants were present at the same nucleotide in *trans,* yielding the Sanger sequencing interpretation complex. WGS allowed for easier identification by separation of the two alleles.

Another patient carried a nonsense mutation and a splice mutation in *GJB2* (patient P3). In routine care in 2000s, splice sites were not screened, and this splice variation (c.-23 + 1G > A) was described for the first time in 2002 [1]. This can explain why we discovered this mutation by WGS analysis. Since 2005, splice sites have been screened in all patients in routine care. Since this case, we screened the splice sites on all DNA from patients before 2005, and another patient with this variant was detected.

For patient P4, the second event (c.269dup) was not detected by DHPLC. Indeed, in the 2000s, for the first screening, DHPLC was used to detect nucleotide alterations in PCR products without sequencing. When a mutation was detected by DHPLC, it was confirmed by Sanger sequencing. However, for this case, DHPLC did not detect the c.269dup variant showing a technical deficit.

After these WGS results, all *DFNB1* monoallelic patients analyzed with DHPLC were screened by Sanger sequencing.

Moreover, for several years, we were only interested in 2% of the coding regions of the genome, the exome [19]. However, recent studies interesting by non-coding DNA, showed that 8% of genome correspond to cCRE regions [22]. Discovery and interpretation of these cCREs are important because they play important roles on gene expression, chromatin organization at different cell states [22].

Many publications have highlighted defects in chromatin organization and the disruption of regulatory regions leading to pathologies, and many publications describe defects in embryonic development or cancers [31,32,33]. Therefore, it is important to consider the non-coding genome and non-coding variations.

Few publications describe *GJB2* gene regulation, at the proximal or distal level. Human *GJB2* promoter has been described for the first time in 1997 by Kiang and his collaborators [34]. The following year, promoter and basal regions essential for *GJB2* expression have been characterized [35]. Following Wilch’s publication hypothesized that some deafness could result from disruption of *GJB2 cis*-regulatory elements (CREs) [16,36], we were the first to explore *GJB2 cis*-regulation. We described several CREs of *GJB2* and suggested a 3D *DFNB1* regulation locus model [30].

Indeed, with this paper, we bring to light the impact of one non-coding variant. Indeed, one patient carries a promoter variation which has an impact on *GJB2* expression. *GJB2* expression is reduced and when this variant is present in conjunction with the *GJB2* enhancer, *GJB2* expression is reduced to 50%. Each allele of patient P10 carries a *GJB2* mutation, the nonsense mutation produces a connexin26 truncated protein and the upstream variation decreases *GJB2* expression. This patient presents mild deafness, which can be explained by *GJB2* transcripts reduced by promoter variants, but not completely.

This is the second mutation reported to affect the *GJB2* promoter. Matos et al., found a g.-77C > T variant in the compound heterozygous state with mutation p.(Val84Met) in a Portuguese deaf patient. This variant is also in the GC box at −81 pb of TSS and alters transcription [37].

After these results, we screened 25 monoallelic other *DFNB1* patients (rare variant or c.35delG), but none of them presented this promoter variant. It is most likely a private variant.

Moreover, this upstream variation may explain deafness phenotypes (degree of hearing loss) [38,39]. Indeed, if *GJB2* expression is altered, the connexin 26 level is modified and modulates the phenotype.

Some cases of presbycusis emerge more prematurely, maybe it is due to decrease of *GJB2* expression by a *cis*-regulatory variants and correlates with environment factors.

These results demonstrate the necessity to investigate the non-coding genome to overcome wandering and diagnostic deadlock with investigation of chromatin conformation and regulatory variants. Functional assays will be necessary to explore the function of non-coding regions and validate new genes or variants of uncertain significance implicated in genetic diseases.

*Cis*-acting regions may be a therapeutic challenge that could lead to the development of specific molecules capable of modulating gene expression in the future. A better understanding of regulatory mechanisms of gene expression could elucidate cases of patients where the phenotype is not yet explained by the genotype. This would thus help in better diagnosis and therefore better management. The analysis of *cis*-regulatory regions will allow the functional impact exploration of certain genomic rearrangements or variants in the chromatin organization. When genomic rearrangements or SNVs are located in non-coding regions, a study of the three-dimensional organization of chromatin study and an analysis of the TADs (Topologically associated domains) in the loci possibly involved in genetic diseases will be proposed by chromatin conformation approaches derived from 3C (Chromosome Conformation Capture). Functional assays (gene reporter tests, chromatin immunoprecipitation and CRISPR-Cas9 analysis) could be used to validate potential regulatory variants.

Six genotypes remained unresolved, although the *DFNB1* locus has been screened along with other deafness genes. This may be explained by the fact that the genome hg38 was not complete, some regions are hidden as repeat sequences or telomeric regions. However, new human reference genome described 2226 paralogous gene copies, whose 115 are predicted to be protein coding, and cover repeat sequences but their function is unknown [40,41]. Perhaps it is not surprising that we cannot resolve all genotypes if we do not have all the information.

Structural variations analysis did not provide answers. Upstream WGS and CGH arrays were performed on *DFNB1* locus for all patients to detect *DFNB1* deletions or duplications. However, CGH arrays hide repeated sequences. Thus, these sequences are excluded. Moreover, CGH arrays cannot identify unbalanced rearrangements. After WGS and SV analysis we hoped to detect unbalanced rearrangements, but the bioinformatic tool, BreakDancer, used to analyze structural variations did not highlight causal variations at the *DFNB1* locus [42,43]. WGS produces a lot of data, but SV detection algorithms have strengths and weaknesses, and some algorithms do not allow the identification of all types of SV [42,44]. We could perhaps discuss whether we did not detect SVs because patients did not have SVs or whether this was due to technology limitations.

Although the *DFNB1* locus was screened, we do not have explanations for these patients. There are some limitations in methodologies which remove repeated sequences or non-aligned reads; it would be interesting to use other techniques such as PacBio Sequencing to have long reads.

Through this project, we aim to better understand molecular mechanisms and we expect to reduce the diagnostic odyssey. Ensuring a definitive diagnosis will have a huge impact on patients who have spent years in a diagnostic deadlock, receiving multiple misdiagnoses resulting in inappropriate treatments.

Furthermore, every test, procedure, treatment, on a misdiagnosis amounts to wasteful spending in healthcare.

Thus, patients and families should receive appropriate therapy with a good genetic counselling, and be well-managed and advised in their daily lives.

## 5. Conclusions

In conclusion, WGS allowed us to correct the genotypes of patients with NSHL by rectifying routine care Sanger sequencing.

We resolved 4 genotypes out of 10 by identifying a second event on the *GJB2* gene, including a promoter variant. Indeed, WGS identified a non-coding variation in the *GJB2* promoter, with an impact on *GJB2* expression.

Six genotypes remain unresolved, amongst which one is possibly due to another deafness gene (*USH1C*).

In this day and age, WGS is an essential tool, as illustrated well in this work, and we propose that it is today to best strategy for improving knowledge for deafness patients.

## Figures and Tables

**Figure 1 genes-12-01267-f001:**
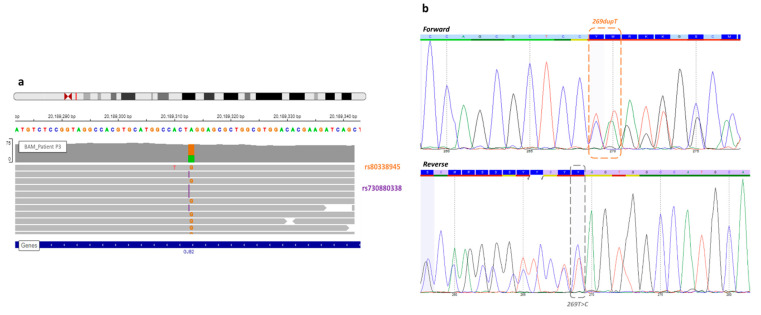
View of the missense variant and frameshift *GJB2* gene of patient P3. (**a**) In the Integrative Genome Viewer, the frameshift variant (rs730880338) at c.269 *GJB2* position, known before WGS (orange), and the missense variant (rs80338945) discovered by WGS analysis (purple). Each variant was on a different read, so this analysis confirmed a *trans* configuration. (**b**) A new Sanger sequencing in forward and reverse detected both mutations, but it still remains difficult to interpret.

**Figure 2 genes-12-01267-f002:**
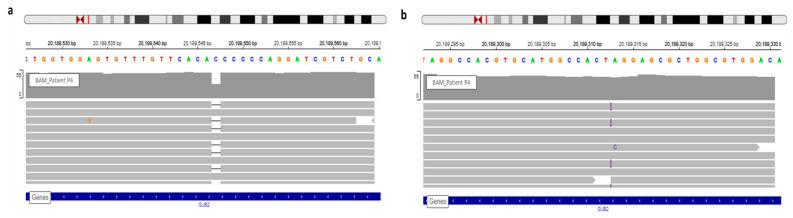
View of *GJB2* variants of patient P4 in IGV. (**a**) WGS confirmed the known c.35delG. (**b**) The frameshift variation, c.269dup, detected via WGS and observed in IGV.

**Figure 3 genes-12-01267-f003:**
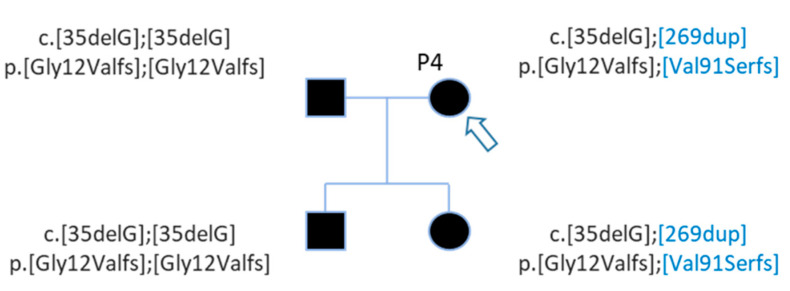
Pedigree of patient P4 with 2 *GJB2* variants. Patient P4 (arrow) carries two *GJB2* mutations, the c.35delG known before WGS and the c.269dup discovered by WGS analysis (in blue). WGS analysis detected mutations for her daughter also.

**Figure 4 genes-12-01267-f004:**
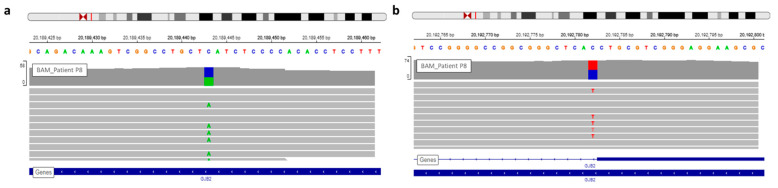
View of *GJB2* variants of patient P8 in IGV. (**a**) The nonsense variation, c.139G > T, discovered during routine care. (**b**) The second *GJB2* variation detected by WGS analysis is a splice site mutation.

**Figure 5 genes-12-01267-f005:**
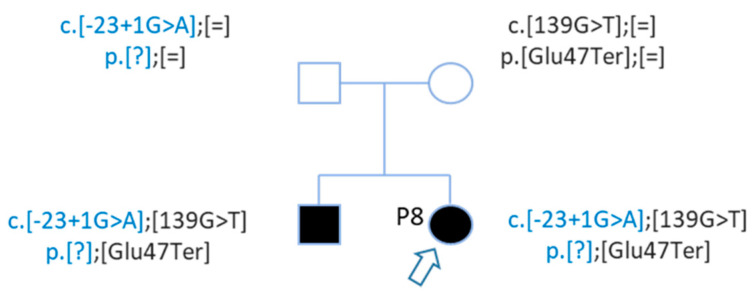
Pedigree of patient P8 with 2 *GJB2* mutations in trans. Patient P8 and her brother carried 2 mutations in trans. Mutations were inherited from each parent.

**Figure 6 genes-12-01267-f006:**
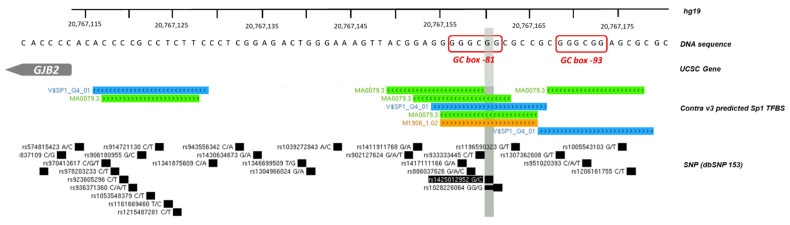
*GJB2* promoter and upstream variants detected via WGS analysis. The *GJB2* promoter has 2 GC boxes (each located respectively at −81 and −93 bp of TSS) that are useful for *GJB2* basal transcription. The upstream *GJB2* variant is located on GC box -81 and Contrat v3 predicts several Sp1 binding sites at this location (Bioinformatic tool to predict transcription factor binding sites http://bioit2.irc.ugent.be/contra/v3/#/step/1, the 21 May 2021).

**Figure 7 genes-12-01267-f007:**
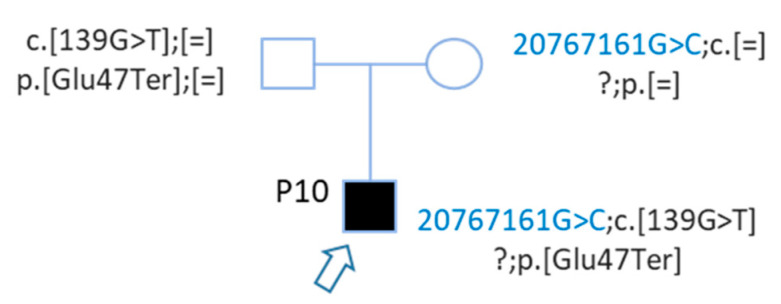
Segregation of *GJB2* variants of patient P10. Hearing parents were carrying each a *GJB2* variant, the nonsense and the upstream variant. Patient P10 carried these 2 mutations in trans.

**Figure 8 genes-12-01267-f008:**
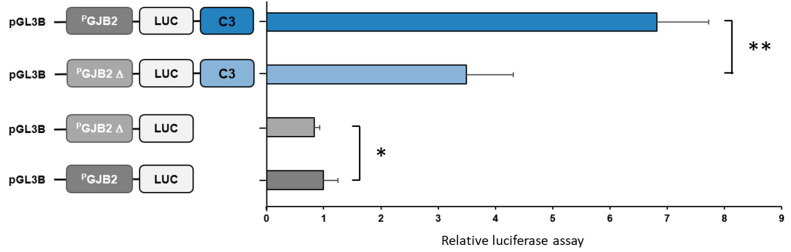
Functional assays of *GJB2* promoter variant. Luciferase reporter constructs with the *GJB2* WT promoter (P*_GJB2_*; 1043 bp), *GJB2* mutated promoter (P_GJB2_ Δ; 1043 bp), and constructions with C3 enhancer were transfected in SAEC cells. * *p* < 0.05; ** *p* < 1.33 × 10^−7^ using analysis of variance and *t*-tests.

**Figure 9 genes-12-01267-f009:**
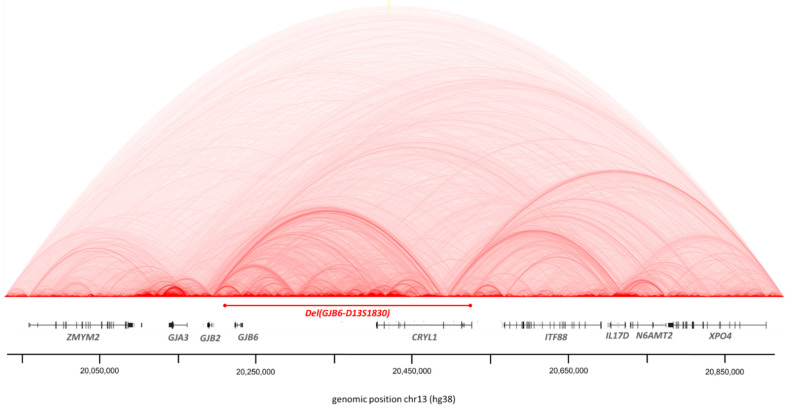
Deletion del(GJB6-D13S1830) of patient P5. Deletion del(GJB6-D13S1830) disrupted *GJB6* gene and *CRYL1* gene but *GJB2* remained intact.

**Table 1 genes-12-01267-t001:** Patient genotypes (monoallelic) at the *DFNB1* locus before WGS.

Patient	Deafness Phenotype	Case	Gene	HGVSc	Chr.	Position (hg38)	HGVSp	Impact	Consequences	Variant Class	rs Number	Allele Frequency (GnomAD)
P1	Profound	Single	*GJB2*	NM_004004.5:c.132G > A	chr13	20189450	NP_003995.2:p.Trp44Ter	HIGH	stop_gained	SNV	rs104894407	0.00001470
P2	Profound	Single	*GJB2*	NM_004004.5:c.169C > T	chr13	20189413	NP_003995.2:p.Gln57Ter	HIGH	stop_gained	SNV	rs111033297	0.00005879
P3	Profound	Single	*GJB2*	NM_004004.5:c.269dup	chr13	20189312	NP_003995.2:p.Val91SerfsTer11	HIGH	frameshift_variant	insertion	rs730880338	0.00002940
P4	Profound	Family	*GJB2*	NM_004004.5:c.35del	chr13	20189546	NP_003995.2:p.Gly12ValfsTer2	HIGH	frameshift_variant	deletion	rs80338939	0.009802
P5	Profound	Single	del(*GJB6*-D13S1830)									0.0003935
P6	Profound	Single	*GJB2*	NM_004004.5:c.169C > T	chr13	20189413	NP_003995.2:p.Gln57Ter	HIGH	stop_gained	SNV	rs111033297	0.00005879
P7	Profound	Single	*GJB2*	NM_004004.5:c.633T > A	chr13	20188949	NP_003995.2:p.Cys211Ter	HIGH	stop_gained	SNV	-	-
P8	Profound	Family	*GJB2*	NM_004004.5:c.139G > T	chr13	20189443	NP_003995.2:p.Glu47Ter	HIGH	stop_gained	SNV	rs104894398	0.0001176
P9	Profound	Single	*GJB2*	NM_004004.5:c.313_326del	chr13	20189255	NP_003995.2:p.Lys105GlyfsTer5	HIGH	frameshift_variant	deletion	rs111033253	0.0003234
P10	Mild	Single	*GJB2*	NM_004004.5:c.139G > T	chr13	20189443	NP_003995.2:p.Glu47Ter	HIGH	stop_gained	SNV	rs104894398	0.0001176

**Table 2 genes-12-01267-t002:** *DFNB1* mutations detected by WGS.

Patient	Gene	HGVSc	Chr.	Position (hg38)	HGVSp	Impact	Consequences	Variant Class	rs Number	Allele Frequency (GnomAD)
P3	*GJB2*	NM_004004.5:c.269dup	chr13	20189312	NP_003995.2:p.Val91SerfsTer11	HIGH	frameshift_variant	insertion	rs730880338	0.00002940
	*GJB2*	NM_004004.5:c.269T > C	chr13	20189313	NP_003995.2:p.Leu90Pro	MODERATE	missense_variant	SNV	rs80338945	0.001161
P4	*GJB2*	NM_004004.5:c.35del	chr13	20189546	NP_003995.2:p.Gly12ValfsTer2	HIGH	frameshift_variant	deletion	rs80338939	0.009802
	*GJB2*	NM_004004.5:c.269dup	chr13	20189312	NP_003995.2:p.Val91SerfsTer11	HIGH	frameshift_variant	insertion	rs730880338	0.00002940
P8	*GJB2*	NM_004004.5:c.139G > T	chr13	20189443	NP_003995.2:p.Glu47Ter	HIGH	stop_gained	SNV	rs104894398	0.0001176
	*GJB2*	NM_004004.5:c.-23 + 1G > A	chr13	20192782	.	HIGH	splice_donor_variant	SNV	rs80338940	0.0003236
P10	*GJB2*	NM_004004.5:c.139G > T	chr13	20189443	NP_003995.2:p.Glu47Ter	HIGH	stop_gained	SNV	rs104894398	0.0001176
	*GJB2*	G > C	chr13	20193022	-	UNKNOWN	upstream_gene_variant	SNV	rs1425012952	0.00001472
	*GJB2*	G > T	chr13	20183294	-	UNKNOWN	downstream_gene_variant	SNV	rs372782198	0.003099

**Table 3 genes-12-01267-t003:** USH1C variants of patient P5. Three variants detected via WGS analysis.

Gene	HGVSc	Chr.	Position (hg38)	Protein Variation	Impact	Consequences	Variant Class	rs Number	Allele Frequency (GnomAD)
*USH1C*	NM_001297764.1:c.463C > T	chr11	17527256	R155 *	HIGH	stop_gained	SNV	rs377145777	0.00001759
*USH1C*	NM_001297764.1:c.1589 + 3_1589 + 6del	chr11	17496751	...	LOW	splice_region_variant&intron_variant	deletion	-	-
*USH1C*	NM_005709.3:c.1557G > C	chr11	17498195	E519D	MODERATE	missense_variant	SNV	rs1064074	0.5511

* Stop codon.

## Data Availability

Data are available upon request from the authors.

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
