# Peer review of "Whole-Genome Sequencing Improves the Diagnosis of DFNB1 Monoallelic Patients"

_genes, 2021, doi:10.3390/genes12081267_

Round 1

Reviewer 1 Report

Comments and Suggestions for Authors

Anaïs Le Nabec and colleagues presented the manuscript "Whole genome sequencing improves diagnosis of DFNB1 monoallelic patients" in which they conducted a search of pathogenic variants leading to hearing loss that were not detected during routine screening. A definite problem in molecular diagnostics of hearing loss is the identification of only one mutant recessive allele in the GJB2 gene in patients with hearing impairments. When searching for previously unidentified variants in 10 patients with monoallelic mutations, the authors used whole genome sequencing (WGS). Based on the data obtained, the authors were able to find additional variants in 4 out of 10 patients, including one previously unexplored variant.

I have some comments which I hope will help improve the manuscript.

Introduction

- Page 1, lines 33-36:  “… and other have a mitochondrial inheritance mode (DFNM) [5,6].” – It is wrong because according to Hereditary Hearing Loss Homepage. https://hereditary hearing loss.org, the designation DFNM is intended for the modifier loci, not mitochondrial loci.

- Page 1, line 41: the title of the GJB2 gene according to NCBI is “gap junction protein beta 2” and it is better to add its chromosomal location.

- Page 1, line 45: del-101 kb (GJB2-D13S175) – this deletion is named del(GJB2-D13S175) according to reference [13].

Materials and Methods / 2.6. Functional assays cells

  • What cells were used? - Information is missing.

Results

  • It will be better to specify the deafness phenotype (mild / moderate / severe / profound, and the age of onset, single or family cases) for each of patients, for example, by adding this information in Table 1.

  • WGS was applied to all 10 patients, although it seems that it would be easier, before WGS, to carry out Sanger sequencing of exon 1 (with adjoined intron region) and coding exon 2. Moreover, on page 2 (lines 85-88) it was said that “Before WGS, Sanger sequencing had been realized to screen exon 2, intron 1 and UTR sequences of the GJB2” However, with regard to patient 4: “… a recurrent mutation, in deaf population, the c.35delG (rs8033893) detected by DHPLC (Denaturing High Performance Liquid Chromatography)…”. It was unclear whether the DHPLC results were confirmed by Sanger sequencing in both (forward and reverse) directions? In this case, the second mutation, c.269dup, would have been detected by Sanger sequencing of exon 2 in reverse direction and without WGS.

The same situation with patient 8 - Sanger sequencing of exon 1 (with adjoined intron region) in this patient before WGS would allow to make a diagnosis.

  • Figure 6 (Legend): “…Contrat v3 predicts several Sp1 binding sites at this location” - what is the tool “Contrat v3” ?

  • Page 8, lines 233-234: “Moreover, in our previous paper, the C3 region was described for the first time as the most important GJB2 enhancer [30].” – Please add more information (size, localization) about the C3 region.

Discussion

  • Information in two paragraphs (page 11 lines 294-299, 300-304) contradicts “Before WGS, Sanger sequencing had been realized to screen exon 2, intron 1 and UTR sequences of the GJB2” (Material and Methods, page 2, lines 85-88).

Minor comments

  • The gene variants should be uniformly named according to the accepted guidelines, for exampleGlu47Ter instead of p.Glu47*; c.35delG instead of 35delG, etc. This comment also applies to the designation of genotypes. The authors should check the designations of all variants and genotypes throughout the text and all figures.

  • The manuscript needs proofreading, it is not free from typos and errors, for example, several of them:

  • Page 4, line 156: “a heterozygote deletion del(GJB6-D13S1830) “- a heterozygous deletion del(GJB6-D13S1830)
  • Page 5, line 167: “the GJB2 gene of patient P3allowed” – “the GJB2 gene of patient P3 allowed”
  • Page 9, line 253: “but GJB2remained intact” – “but GJB2 remained intact”
  • Page 11, line 278: “These SV dodn’t disrupt either the GJB2 gene or the DFNB1 - These SV do not disrupt either the GJB2 or DFNB1 gene
  • Page 12, line 336: “It is most-likely a private variant” – “It is most likely a private variant”
  • Page 12, line 339: “the connexin26 level is modified” – “the connexin 26 level is modified”

Author Response

First of all, thanks to reviewer 1 for the comments and suggestions that I am sure will improve our manuscript.

Point 1: - Page 1, lines 33-36:  “… and other have a mitochondrial inheritance mode (DFNM) [5,6].” – It is wrong because according to Hereditary Hearing Loss Homepage. https://hereditary hearing loss.org, the designation DFNM is intended for the modifier loci, not mitochondrial loci.

Response 1: We agree with the Reviewer, it is a very good remark, and we modified the text accordingly.

Point 2: - Page 1, line 41: the title of the GJB2 gene according to NCBI is “gap junction protein beta 2” and it is better to add its chromosomal location.

Response 2: Thank you, indeed it is more precise with chromosomal location. We add this information on the paper: chr13:20,187,470-20,192,938 (hg38).

 Point 3: - Page 1, line 45: del-101 kb (GJB2-D13S175) – this deletion is named del(GJB2-D13S175) according to reference [13].

Response 3: We appreciate the Reviewer advice, it is absolutely right, it is a mistake on our part.

Point 4: What cells were used? - Information is missing.

Response 4: We are sorry to miss this information, the cells we have used: SAEC: Small Airways Epithelial cells.

Point 5: It will be better to specify the deafness phenotype (mild / moderate / severe / profound, and the age of onset, single or family cases) for each of patients, for example, by adding this information in Table 1.

Response 5: We understand the Reviewer and we added informations about patients on the Table 1. 

Point 6: WGS was applied to all 10 patients, although it seems that it would be easier, before WGS, to carry out Sanger sequencing of exon 1 (with adjoined intron region) and coding exon 2. Moreover, on page 2 (lines 85-88) it was said that “Before WGS, Sanger sequencing had been realized to screen exon 2, intron 1 and UTR sequences of the GJB2” However, with regard to patient 4: “… a recurrent mutation, in deaf population, the c.35delG (rs8033893) detected by DHPLC (Denaturing High Performance Liquid Chromatography)”. It was unclear whether the DHPLC results were confirmed by Sanger sequencing in both (forward and reverse) directions? In this case, the second mutation, c.269dup, would have been detected by Sanger sequencing of exon 2 in reverse direction and without WGS.

The same situation with patient 8 - Sanger sequencing of exon 1 (with adjoined intron region) in this patient before WGS would allow to make a diagnosis.

Response 6: We understand the Reviewer comment and we agree with his comment. It would have been much better if we had realized a Sanger sequencing of all the GJB2 gene before WGS. Our routine lab, we used DHPLC to screen exon 2 of GJB2, since no hetero and homo-duplex detected we did not realize Sanger sequencing. In this case, DHPLC did not detect c.269dup mutation, this is why genotype of patient 4 was not resolved before WGS. Indeed, if we have realized a Sanger sequencing in sense and reverse direction would detect the mutation. It was what we did in the past, 15 years ago. We modified the text to be more clear.

And for patient 8, this is a reflection of the situation in 2002, where we did not screen the exon 1. We agree with Reviewer comment and it would have been more appropriate to screen exon 1 before WGS. Nowadays, it is the strategy which have now been in place in our laboratory.

Point 7: Figure 6 (Legend): “…Contrat v3 predicts several Sp1 binding sites at this location” - what is the tool “Contrat v3” ?

Response 7: Thanks to Reviewer for this comment. Contra V3 is a bioinformatic tool (http://bioit2.irc.ugent.be/contra/v3/#/step/1) which allows easy visualization and exploration of predicted transcription factor binding sites (TFBSs) in any genomic region surrounding coding or non-coding genes. We add this information in Materials and Methods in the manuscript.

Point 8: Page 8, lines 233-234: “Moreover, in our previous paper, the C3 region was described for the first time as the most important GJB2 enhancer [30].” – Please add more information (size, localization) about the C3 region.

Response 8: We appreciate the Reviewer advice. We add this information about C3 region on the manuscript: chr13:20419404-20420446 (hg38) – 1043pb.

Point 9: Information in two paragraphs (page 11 lines 294-299, 300-304) contradicts “Before WGS, Sanger sequencing had been realized to screen exon 2, intron 1 and UTR sequences of the GJB2” (Material and Methods, page 2, lines 85-88).

Response 9: We thank the Reviewer to detect this contradiction between these two sentences. We correct the sentence in Material and Methods, which it is wrong.

Point 10: The gene variants should be uniformly named according to the accepted guidelines, for exampleGlu47Ter instead of p.Glu47*; c.35delG instead of 35delG, etc. This comment also applies to the designation of genotypes. The authors should check the designations of all variants and genotypes throughout the text and all figures.

Response 10: We are sorry for these errors, we corrected the gene variant names according to the accepted guidelines. 

Point 11:

  • The manuscript needs proofreading, it is not free from typos and errors, for example, several of them:
  • Page 4, line 156: “a heterozygote deletion del(GJB6-D13S1830) “- a heterozygous deletion del(GJB6-D13S1830)
  • Page 5, line 167: “the GJB2gene of patient P3allowed” – “the GJB2 gene of patient P3 allowed”
  • Page 9, line 253: “but GJB2remained intact” – “but GJB2remained intact”
  • Page 11, line 278: “These SV dodn’t disrupt either the GJB2gene or the DFNB1 - These SV do not disrupt either the GJB2 or DFNB1 gene
  • Page 12, line 336: “It is most-likely a private variant” – “It is most likely a private variant”
  • Page 12, line 339: “the connexin26 level is modified” – “the connexin 26 level is modified”

Response 11: We are sorry for typos and errors. We thank the Reviewer to highlight it, we correct the manuscript.

Reviewer 2 Report

The paper is overall well written and the data are presented clearly. 
I have only some suggestions about the form:

“This type of deafness affects all 39 frequencies and is not associated with inner ear malformation and vestibular 40 function remains unchanged” could be switched to “This type of deafness affects all 39 frequencies and it is not associated with inner ear malformationS, while vestibular  function remains unaffected

“populations with the most  frequent mutation….”

“GJB2 expression is divided by two” : reduced to 50%

Other considerations:

1) As I can see, several of the patients with monoallelic mutation had been analyzed by Sanger. Considering that a lot of diagnostic laboratories have in place Whole exome sequencing and not WGS, it could be useful to specify which mutations in this paper could have been detected by both WES and WGS and which only by WGS.

2) Array has been correctly performed in order to detect eventual CNVs. Small deletions could not be revealed by Array, but only by MLPA. Which is the detection rate for small deletions with the WGS available in your laboratory?

3) Finally, the authors state: “the DFNB1 locus has been screened 362 along with other deafness genes”. Which panel has been analyzed? How many genes have been sequenced?

Author Response

First of all, thanks to reviewer 2 for the comments and suggestions that I am sure will improve our manuscript.

Point 1: “This type of deafness affects all 39 frequencies and is not associated with inner ear malformation and vestibular 40 function remains unchanged” could be switched to “This type of deafness affects all 39 frequencies and it is not associated with inner ear malformationS, while vestibular function remains unaffected

 “populations with the most  frequent mutation….”

“GJB2 expression is divided by two” : reduced to 50%

Response 1: We are sorry for typos and errors. We thank the Reviewer to highlight it, we corrected the manuscript.

Point 2: 1) As I can see, several of the patients with monoallelic mutation had been analyzed by Sanger. Considering that a lot of diagnostic laboratories have in place Whole exome sequencing and not WGS, it could be useful to specify which mutations in this paper could have been detected by both WES and WGS and which only by WGS.

Response 2: Thanks to the Reviewer for his comment. Indeed, several mutations could have be detected by WES as c.269T>C variant (P3), c.269dup (P4) or the variant c.-23+1G>A (P8). However, WES would not have detected GJB2 promoter variant (P10). Moreover, we started this project without prejudice so WGS allowed to better respond to this answer by studying of non-coding and intronic DNA. WGS allows the detection of small deletion in the non-coding regions of the genome as well as some inversion that are not detected by Exome sequencing. We added a sentences in the discussion about WES and WGS.

Point 3: Array has been correctly performed in order to detect eventual CNVs. Small deletions could not be revealed by Array, but only by MLPA. Which is the detection rate for small deletions with the WGS available in your laboratory?

Response 3: Here, WGS allows to detect very small deletions (few hundred base pair, sometimes less). We added a sentence in the Discussion in the manuscript.

Point 4: Finally, the authors state: “the DFNB1 locus has been screened 362 along with other deafness genes”. Which panel has been analyzed? How many genes have been sequenced?

Response 4: WGS allowed to screen all genes but we focused on 215 deafness genes. We added these genes on Supplementary Material.
